# A Critical Analysis of Cooperative Caching in Ad Hoc Wireless Communication Technologies: Current Challenges and Future Directions

**DOI:** 10.3390/s25041258

**Published:** 2025-02-19

**Authors:** Muhammad Ali Naeem, Rehmat Ullah, Sushank Chudhary, Yahui Meng

**Affiliations:** 1School of Science, Guangdong University of Petrochemical Technology, Maoming 525000, China; malinaeem7@gmail.com; 2School of Computing, Newcastle University, Newcastle upon Tyne NE4 5TG, UK; rehmat.ullah@ncl.ac.uk; 3School of Computer, Guangdong University of Petrochemical Technology, Maoming 525000, China; sushankchudhary@gmail.com

**Keywords:** wireless network, caching, wireless sensor network

## Abstract

The exponential growth of wireless traffic has imposed new technical challenges on the Internet and defined new approaches to dealing with its intensive use. Caching, especially cooperative caching, has become a revolutionary paradigm shift to advance environments based on wireless technologies to enable efficient data distribution and support the mobility, scalability, and manageability of wireless networks. Mobile ad hoc networks (MANETs), wireless mesh networks (WMNs), Wireless Sensor Networks (WSNs), and Vehicular ad hoc Networks (VANETs) have adopted caching practices to overcome these hurdles progressively. In this paper, we discuss the problems and issues in the current wireless ad hoc paradigms as well as spotlight versatile cooperative caching as the potential solution to the increasing complications in ad hoc networks. We classify and discuss multiple cooperative caching schemes in distinct wireless communication contexts and highlight the advantages of applicability. Moreover, we identify research directions to further study and enhance caching mechanisms concerning new challenges in wireless networks. This extensive review offers useful findings on the design of sound caching strategies in the pursuit of enhancing next-generation wireless networks.

## 1. Introduction

The Internet was created with the use case of file sharing in mind, and later on, researchers started enhancing the Internet model to handle real-time traffic, such as video streaming. Nowadays, the Internet is used mostly for the dissemination of popular content. As a result, data traffic is increasing day by day. In the year 2000, file-sharing traffic accounted for roughly 80% of the overall Internet traffic [1]. Today, video streaming (e.g., YouTube) traffic accounts for about 86% of global Internet traffic [2]. According to the latest forecast, approximately 34.7 million messages are transmitted on the current Internet every minute. Google has indexed more than 130 trillion web pages [3]. Every minute, 1.2 petabytes of traffic are communicated through the Internet, while 10 million videos are watched on mobile phones [4]. 

To alleviate the pressure put on the Internet by such heavy usage and help scale content dissemination, the community has proposed several solutions based on caching popular content close to the users to reduce the content access times and the generated traffic [5,6]. At the same time, wireless communication comes with its challenges. For instance, it is difficult to choose an interface path to manage data transmission and connections among the different mobile hosts. In addition, congestion is getting out of control, and new mechanisms are needed to fulfill the wireless-based network quality requirements [7]. Therefore, several wireless technologies have been designed to overcome these limitations, such as wireless sensor networks (WSNs), vehicular networks, ad hoc networks, and wireless mesh networks (WMNs). 

In all these approaches, caching is a central component that is able to enhance the overall performance. Caching is made up of two methods: cooperative caching and non-cooperative caching [8]. Cooperative caching is distributed among the network nodes and allows the caches of the disseminated contents to fulfill the future end-users’ requirements. Therefore, the users can fetch the cached contents from caching nodes [9]. This reduces the load on the network and alleviates the uncontrolled congestion across the network. Moreover, cooperative caching among the network nodes has several performance benefits. For instance, it allows users to potentially retrieve content from the nodes nearby as opposed to from the gateway. As a result, the throughput is improved since it decreases rapidly with an increase in hop count in a multi-hop network. In addition, cooperative caching enhances the local communication capacity of wireless links. Furthermore, if multiple copies of requested content are cached at different locations, cooperative caching facilitates users to choose the best option to fetch the content from the appropriate node with high-quality throughput [10,11,12].

A massive amount of applications is running over the wireless Internet, which significantly requires optimization for communication among the network nodes to serve users’ requests with minimal consumption of energy and short delays [13]. For instance, the battery life of wireless-based devices will be extended if the amount of communicated content is minimized. Therefore, this change can be implemented by caching useful content at each node, either in their local store or at the nearest neighboring nodes. Furthermore, caching is effective in minimizing the requirements of a wide network. Therefore, cooperative caching is a promising approach to meeting these goals with high efficiency [14,15]. Cooperative caching in wireless networks significantly minimizes service latency by caching popular content at local base stations because there is no need to retrieve content through backhaul. To this end, in this study, we survey different caching approaches and designs in wireless communications, focusing primarily on cooperative caching [16,17,18]. In this study, we make the following contributions:We present the state-of-the-art cooperative caching approaches that have been proposed to improve wireless-based ad hoc network environments.We identify and discuss the key contributions of cooperative caching in wireless-based Internet technologies, such as lower usage of network resources (power and energy), minimum query delay, improved playback, reduced retrieval delay, improved success ratio, reduced buffer utilization, improved data availability, reduced network congestion, and improved connectivity of mobile vehicles.We categorize the studied caching approaches based on their wireless application scenarios, such as mobile ad hoc networks (MANETs), wireless mesh networks (WMNs), wireless sensor networks (WSNs), and vehicular ad hoc networks (VANETs).We outline the present challenges in wireless-based ad hoc network technologies and identify different directions for future research.

In Section 2, related and existing studies are mentioned with their contributions and limitations. In Section 3, caching design concepts are presented, focusing on cooperative and non-cooperative caching designs. In Section 4, the basic role of this study is illustrated, explaining cooperative caching for the diverse kinds of wireless networks. Section 5 consists of a summary and a discussion of wireless communication technologies. Section 6 provides several contributions to address wireless technologies. In Section 7, future research directions to improve the quality of wireless heterogeneous networks are explored. In Section 8, we conclude the paper.

## 2. Background and Existing Surveys

Caching is an integral component of modern technologies designed to improve the quality of data dissemination by caching the content at intermediate locations during their transmission [19]. Caching can be cooperative and non-cooperative, where cooperative caching shares the content caching state information with its neighbor nodes and the content caching decisions are made through the coordination of multiple nodes. If a request for content cannot be satisfied, it is forwarded to the neighboring nodes to download the required content. Cooperative caching among the cache-enabled nodes provides several performance benefits. For instance, in a multi-hop network scenario, it allows users to potentially retrieve content from the nodes nearby as compared to the gateway. As a result, the throughput increases due to a lower hop count. Furthermore, if a massive amount of applications are running over the wireless Internet, the optimization of network resources needs to be taken into account so that communication among the nodes may be efficient enough to serve users’ requests with minimal energy consumption and with short delays [20,21,22]. For instance, the battery life of wireless devices will be extended if the amount of communicated content is minimized. This can be achieved by caching useful content at each node, either at their local store or at the nearest neighboring nodes. In non-cooperative-based caching, all the network nodes decide to cache the data individually, and their cache state is not allowed to be shared with neighbors. Caching coordination is not allowed in this approach because each node has to decide whether to cache the content or not. It should be noted that work on non-cooperative caching is out of the scope of this paper. Recently, Van et al. [23] published a survey paper that discusses in-network caching strategies in Information-Centric Networks (ICNs), including MANET, VANET, IoT, and WSNs, as well as on-path, off-path, content, network, and node caching, along with their trends, issues, and future research. In addition, a study by Obaid et al. [19] provides a systematic review that evaluates caching patterns in ad hoc networks, concentrating on the consequences of caching for network performance, possible size, vitality usage, and adaptability to a fluid topology. The article reviews issues that had not been addressed previously and provides potential solutions to improve routing protocols. Moreover, Khalid et al. [24] review caching techniques in wireless ad hoc networks (NDNs), addressing data duplication, mobility, and location-dependency issues. They highlight NDNs’ advantages, such as caching, mobility, scalability, security, and privacy, and discuss future research. However, these studies recognize that they have some shortcomings, including the lack of consideration of how cooperative caching impacts and mechanism may be affected with regard to different ad hoc network types.

The study of caching has produced a substantial body of scientific literature, and to date, many surveys have been published. In this section, we present a summary of existing recent surveys about caching in wireless-based network infrastructures along with their contributions and limitations and compare them with our survey. Although there is rich literature available on caching, we focus only on the recent cooperative caching for wireless communication networks. In the literature, several survey articles have been published. To the best of our knowledge, all these surveys have limited scopes and lack coverage of the particular approaches of wireless-based networks. We present an in-depth survey regarding the imminent challenges of wireless-based network infrastructures and their solutions through caching. We discuss the representative approaches of cooperative caching, considering the challenges that are considered more problematic for wireless-based networks. We present lessons learned and insights into future emerging solutions that can help fulfill future application requirements. Moreover, caching solutions are suggested at the end of this study that may be beneficial to handle future challenges in wireless communication networks. Table 1 shows recent existing surveys concerning their contributions and limitations.

## 3. Caching Design Concept

Caching is an integral component of modern technologies because it is the most flexible approach to improve the quality of data dissemination by caching the contents at intermediate locations during their transmissions [37,38]. It offers the in-network storage, which is the most beneficial in reducing the complications of the receiver-driven content retrieval process. Moreover, caching is extensively and comparatively studied with IP-based communication to check the performance in terms of retrieval latency [39,40]. It also gives an idea about how the latency depends on the path stretch and bandwidth consumption [41]. The bandwidth is dependent on the way to share among the parallel downloads. The bandwidth can be clarified as the term of capacity that shows the link volume used by the clients’ requests and data contents at a particular point in one second or the capacity of the link to transfer the disseminated requests and data packets at a specific location per unit of time. The latency is increased due to high bandwidth consumption and long stretch paths because in both conditions the client’s requirements are increasing delay. Consequently, if the network congestion is high and the capacity of bandwidth is low, the response latency will be increased. Bandwidth is the amount of data traffic disseminating through a network path or link in a particular time [42,43]. According to the caching design concept, it is divided into two methods, such as cooperative and non-cooperative-based content caching [44].

### 3.1. Cooperative Caching

In cooperative caching, the content caching state is shared with its neighbor nodes, and the content caching decision is taken by using the coordination of multiple nodes. If a request does not find the required content, the request is forwarded to the neighbor nodes to download the appropriate content [8,45]. In Figure 1, the users send their requests to download content C1, C2, and C5. The content C1, C2, and C5 is already published by Publisher 1. Consequently, the requested content C1, C2, and C5 is sent back to the users, and a copy of that content is cached at intermediate routers R1, R2, and R4. After the caching operation, the cache states of all the routers are shared with the neighbors as shown by the information tables of all routers. In Figure 1, the block (a) shows the contents are cooperatively cached at routers R1, R2, and R4. 

### 3.2. Non-Cooperative Caching

In non-cooperative-based caching, all the network nodes are decided to cache the data individually, and their cache state is not allowed to share with neighbors. This approach does not allow caching coordination because each node has to decide whether to cache the content or not [10,46]. Suppose an example to explain non-cooperative-based caching can be defined as when a cache hit takes place, the content is transmitted to the requested user. The required content is cached at the downstream router, and the requested content is deleted from the source to reduce the redundant elements and make room for new content. In this caching, the contents are gradually forwarded towards end nodes near the users to be cached to fulfill the subsequent requests. Figure 1 illustrates the entire process of non-cooperative caching. In this figure, the user requests to download Content C4 from router R5. As the request reaches R5, it finds the requested content there. According to non-cooperative caching, the content is transmitted to the user, and a copy of the requested content is cached at the downstream router (i.e., R2), but the cache of R2 is full. Therefore, the router R2 decided to cache the content C4 individually by deleting some other content. However, the cache state of R2 is not shared with its neighbor nodes (router R5 and R3). Hence, the routers R2, R3, and R5 are non-cooperatively caching the contents as shown by Block (b) in Figure 1.

## 4. Impact of Cooperative Caching in Ad Hoc Networks

The incredible growth of today’s Internet traffic and frequent transmission of similar content generate several problems in which bandwidth utilization, high usage of resources, and power consumption are most important [47]. These problems are hard to resolve by using the present Internet paradigm. Therefore, it needs to change the design of Internet architecture because the consumer is only interested in retrieving their preferred contents rather than the physical locations of hosts. Moreover, the current Internet offers a peer-to-peer communications system that is insufficient to handle the growing issues of arranging effective content dissemination. In this situation, in-network cache storage is deployed to enhance the content dissemination process by caching the disseminated contents at intermediate locations for subsequent content routing [48]. In this way, caching decreases the overall network traffic and delay. Caching is one of the fundamental modules of all modern technologies that are used to accommodate the disseminated content near the users to enhance the content distribution process. Moreover, it decreases the propagation delay and improves the performance of the overall network. Caching delivers several benefits for Internet technologies in which fog computing, edge computing, IoT, and 5G mostly use caching [49]. However, the production of smart devices and multimedia application services such as Video on Demand (VoD) and social networks is continuously increasing video traffic across the Internet. However, the users are interested in retrieving their desired data rather than downloading it from the original servers [50]. To improve data dissemination in wireless networks, efficient data delivery techniques needed to be implemented using a caching module [51]. Recently, a number of methods were proposed to reduce redundant data dissemination in backhaul links. However, cooperative caching is a promising technique to enhance the data transmission efficiency, energy efficiency, and spectrum efficiency at hand [52]. An ad hoc network is a distributed sort of wireless network, and it does not depend on a pre-existing infrastructure such as access points in infrastructure or managed wireless networks. In an ad hoc network, each node contributes to disseminating data dynamically based on connectivity for the other nodes using some routing algorithms. Mobile Ad hoc Network (MANET) [53], Wireless Mesh Network (WMN) [54], Wireless Sensor Network (WSN) [55], and Vehicular Ad hoc Network (VANET) [56] are the most significant applications broadly used to provide communication services. The following sub-sections provide cooperative caching contributions in ad hoc-based wireless networks.

In the following sections, cooperative-based caching in ad hoc networks is thoroughly presented.

### 4.1. Cooperative Caching in Mobile Ad Hoc Network

Mobile ad hoc Network (MANET) [57] is a promising technique that has received a lot of interest due to its flexible approaches in outdoor assemblies, disaster recovery, and battlefields. However, MANET is suffering from problems related to energy consumption, mobility, and route detection. Moreover, the regular usage of route detection is an expensive way that produces insufficient outcomes in terms of Quality of Service (QoS) parameters [58]. In these situations, cooperative caching can be the most flexible approach to provide improved communication services in MANETs, save energy and bandwidth, and reduce data retrieval latency [19].

Caching allows coordination and sharing in which the cached data can be accessed from multiple nodes to fulfill their needs [59]. A protocol named Internet Cache Protocol (ICP) was developed by Claffy and Wessels, which is broadly used to enhance ad-hoc-based communication. In MANETs, the caching protocols are standardized as router-based and message-based protocols. A router-based protocol includes summary cache and cache digest that are used to enable proxies in caching to interchange the information about cached contents. However, the router-based protocol disseminates the incoming users’ requests among a cache array. Consequently, to enhance the overall communication in MANET, some caching schemes were introduced, such as a caching strategy proposed by Satyanarayana et al. [60] in 2022. In this strategy, updating caching with a route update algorithm is proposed to enhance the MANET performance. In this method, the appropriate route to the caching node is determined and shared with the adjacent nodes, and thus all adjacent nodes update their cache by the appropriate route. This caching strategy improves the performance of MANET in terms of packet delivery ratio, packet loss, end-to-end delay, and energy consumption. Moreover, to improve the data access rate in MANET, Sheeba et al. [61] developed a Hybrid cache Management (HCM) strategy where the verification algorithm combines the neighboring contents to provide a spatial query solution. The HCM combines with the cache path and cache data to improve the performance of ad hoc networks by providing enhanced broadcast queries. Caching performance is enhanced using the cache replacement policies. In addition, Islam et al. [62] proposed a cooperative caching strategy for MANET by defining the asymmetric cooperative cache approach for the caching of transmitted data items. Based on the scaled power community index, the network node is selected for the caching of transmitted data items to fulfill the end-user future demands and improve the overall network performance.

Figure 2 illustrates the cacheable MANET in which some devices are equipped with caching capabilities to store the most frequently requested data items that are used to fulfill the subsequent requirements.

### 4.2. Cooperative Caching in Wireless Mesh Network

According to the growing demands of users in wireless networks, the quality of service and network connectivity need new technological approaches to enhance the architecture for ease of deployment at a low cost. A new technique named Wireless Mesh Network (WMN) was proposed to extend the network connectivity and coverage that can provide multi-hop communication. Besides, it opens the network access and delivers high-bandwidth communication [63]. However, WMN is also facing some critical challenges, such as limited resources for device portability and wireless communications. Moreover, data access efficiency is one of the crucial issues that create difficulties in the development of WMN. In such situations, cooperative caching plays a significant role in overcoming these issues of WMN since it provides the facilities to accommodate the frequently requested data items at intermediate locations to access these most desirable items at a lower cost [64]. Moreover, the locally cached copy of a data item can be transmitted to multi-hop communication paths. Therefore, in a caching wireless network, the data can be cached at intermediate nodes or requesting nodes, or it can be cached at neighbor nodes. As a result, the cost of transmissions and usage of resources are minimized [65].

To improve the performance of WMN, a cooperative-based caching technique named Cooperative Approach to Cache Consistency (CACC) [66] was developed to integrate the push and pull-based data dissemination using the hierarchical-based mesh network. Two approaches were introduced in this architecture, in which the first one is used to make efficient cooperation among the network nodes to deliver in-validation reports. The second approach is cooperative invalidation report re-sending, in which the intermediate nodes are used to re-send the missed invalidation reports upon request. As a result, the overall message cost and delay are reduced, and the request arrival rate has been improved in a mesh network. Similarly, CacheRescue [67], another cooperative caching-based technique, was developed to enhance the cache-hit ratio and average data retrieval delay. Advanced Cache Re-transmission Mechanism (ACRM) is proposed by Kuila et al. [68] to improve the performance of WMN. It includes Trend Jacobson and the Cache Retransmission algorithm to enhance the WMN system using relay node caching and interaction logic. The ACRM improves the WMN performance regarding network stability and data transmission efficiency. 

Figure 3 demonstrates the cacheable mesh network environment where network users can directly download their desired data items from the caching nodes to save network bandwidth, resources, and power consumption.

### 4.3. Cooperative Caching in Wireless Sensors Networks

Currently, a considerable amount of content is produced continuously from the sensors of smart devices, and wearable smart devices are producing a large amount of data continuously. Therefore, WSNs have achieved considerable attention for real-time applications due to their wide usage in diverse fields such as disaster management, surveillance systems, health care, and environmental monitoring. Generally, a different number of sensor nodes is deployed randomly in a particular domain to build a WSN [69]. These sensors are used to collect information about the specific area, process it, and send the processed information toward the base stations using single-hop or multi-hop communication. The sensors are associated with a power unit, sensing component, data processing environment, and transmitting module. Power consumption is the most crucial issue for extended processing operations in WSNs because the sensors are equipped with small batteries that exhaust their energy quickly in harsh environments. Several studies have been devoted to solving the energy issue of the sensors, such as duty cycling, energy-aware medium access control, low power communication, and hardware layer protocol [70]. Therefore, energy efficiency is the most noticeable problem in WSNs.

To this end, cooperative caching can play an essential role in these situations to provide an energy-efficient environment for WSNs. In a cooperative caching environment, the transmitted data is cached at different nodes to fulfill future user requirements. Therefore, if a source node is in sleeping mode, the data can be downloaded from the caching node. Moreover, there is no need to transmit multiple requests for similar data to download from the remote source. The sensor nodes sense the environment and cache data items within small storage for a particular time span and then forward the data toward the sink node in WSNs. Besides, the subsequent requests are accomplished from the nearest cached data with less latency. Hence, caching has the ability to improve the overall WSN performance in terms of energy, delay, and network congestion [71]. Recently, Chen-Nakesavula et al. [72] proposed a caching strategy for WSN using the IEEE 802.25.4/ZigBee network. It combines the cooperative cache and Distributed Energy-Aware Routing (DEAR) protocol to determine the request generated by sink nodes and the required data items already available in a local cache. DEAR selects the battery level as a key factor to determine the nodes for data caching at the routing path. WSN is made up of IEEE802.25.4-based sensors and one-hop neighbor nodes. The evaluation is done in NS2, and the proposed caching strategy performs well in terms of byte-hit ratio and energy consumption. Dinh et al. [73] proposed a caching strategy named One Step Ahead (OSA) in which single-server single-cache and multi-server multi-cache are considered to identify the linear combination to cache the data items and fulfill the future users’ demands. Recently, Yang et al. [74] proposed a traffic-based caching mechanism (TCM) in which the popular data items are cached at the nearby sink nodes. More-over, it pushes the popular data items to be cached within the network. The TCM minimizes the number of forwarding requests in multi-hop WSN and reduces the stretch, energy consumption, and data retrieval time. In a study by Yang et al. [74], they present a multi-hop cooperative caching (MCC) and provide a green cooperation scheme for the sensors to identify cooperation among the nodes and the operation scope for the caching decision. The MCC reduces the delay and energy consumption and saves energy loss during data dissemination.

Figure 4 illustrates the cooperative-based cache-able WSN environment in which some of the sensors are equipped with cache memory that usually stores the transmitted data items near the end users to fulfill the subsequent end-users’ requriements.

### 4.4. Cooperative Caching in Vehicular Ad Hoc Network

Vehicular Ad hoc Network (VANET) is made up of a number of connected high-speed vehicles that are associated with communication services. It delivers inter-vehicle communication with the help of multi-hop data dissemination without having any connection to a fixed location [75]. In order to provide communication services such as accessibility and flexible connectivity, it is necessary to integrate vehicular networks with wireless networks, such as local areas or wide-area networks [76]. In the near future, it is expected that the VANET will become a ubiquitous global communication infrastructure. With respect to cellular networks, the VANET includes more heterogeneous applications and has high Quality of Service (QoS) requirements with discrete features [77]. Therefore, context-based services are required for intelligent driving in which the traffic flow, available slots for parking, mobility of surroundings, and the distance from the present position to the next highways need to be found. In these situations, time is a critical entity to be considered because the information changes in each interval due to the continuous change in the driving environment [78]. Therefore, these sorts of features have imposed great challenges on the vehicular network environments. Another critical issue of VANET is the efficient utilization of diversified resources to meet the multi-dimensional QoS requirements of mobile users [79]. Consequently, several cooperative-based caching schemes were developed to enhance data dissemination in VANET. In a study by Gupta et al. [80], a cooperative caching strategy is proposed by combining the Consistent Hash Mobility Prediction (CHMP) algorithm for Vehicular Content-Centric Network (VCCN). Based on the vehicles with the same trajectory, prediction by Partial Matching (PPM) is adopted to determine the vehicles that will travel on a similar path, their moving speed, and directions, and put them into a cluster. In each cluster, CHMP is distributed among the cooperative nodes to reduce cache redundancy and improve data diversity (data availability). Moreover, a popularity-aware data replacement scheme has emerged with a caching strategy to maximize the overall VANET performance. The opportunistic Network Environment (ONE) simulation platform is selected to evaluate the performance. The proposed strategy performed well in terms of average access delay, cache hit ratio, and average hop count.

In a study by Amadeo et al. [81] a dynamic cooperative caching strategy is proposed based on the popular data items. In this strategy, a dynamic caching mechanism is developed to determine the right caching node based on frequent communication with neighbor nodes to cache the most popular data items that meet the future demands of end users by requesting the cached data items. The basic goal of this study is to improve the overall caching performance in VANET using NDN-based cooperative caching mechanisms. Chen et al. [82] proposed a strategy to cache multimedia data items in vehicular networks. A two-layer hierarchical architecture is used, and the proposed scheme allows the data items to cache at network edges chunk by chunk instead of the whole content. The caching location is determined using the cluster head selection, and the content is selected based on the probability matrix where the content popularity and rating score are calculated. The MATLAB simulation platform is selected to experiment, and the proposed caching strategy performed well in terms of average access delay and hit ratio. Cooperative Caching based on Mobility Prediction (CCMP) was proposed by Zhao et al. [83]. In CCMP, an effective freshness caching decision is implemented to improve the content retrieval delay, success rate, and resource utilization in VANET. VANET is sub-categorized into vehicle-to-vehicle (V2V) wire-less networks, as described in the following section.

Figure 5 illustrates the cache-able vehicular network environment in which the small and macro base stations are integrated with cache memory to store the transmitted popular data items to fulfill future requirements. VANET is sub-categorize into Vehicle-to-Vehicle (V2V) wireless networks as described in the following section.

#### Cooperative Caching in Vehicle to Vehicle Network

Currently, the automobile industry and wireless communication have gained significant interest in improving the VANET in the past few years. VANET consists of vehicle-to-infrastructure (V2I) and V2V communications that use wireless access technology (IEEE 802.11p) [84]. This advancement in wireless communication has been considered efficient for road safety and traffic through the implementation of the Intelligent Transport System (ITS). Currently, V2V data dissemination is becoming a significant technological approach to improve the performance of the intelligent transportation system. Moreover, it can enhance the traffic and smart road management systems as it disseminates critical information such as where the accident happened. Also, it improves the communication services to transmit entertainment and new related information to the passengers [85]. However, the V2V system needs highly reliable and high-speed data communication services to disseminate bulky multimedia data such as photos, songs, and videos among the separate vehicles in a vehicular network.

Therefore, to achieve efficient V2V content distribution, caching plays a strategic role in maximizing the throughput of the intelligent transportation system and traffic system. Likewise, to enhance content dissemination in V2V, caching approaches are used to reduce the path stretch in data retrieval. The reason is that the frequently requested data items are offloaded within the cache of several devices and selected vehicles. As a result, the cached data items are used to fulfill the subsequent requests during the off-peak time to respond to the requests at peak traffic hours. The devices and vehicles can retrieve popular content instead of downloading from wireless base stations [86]. Therefore, the direct downloading of the data items can lead to a decrease in link and traffic load within the backhaul and at the base stations as well. Meanwhile, the cache is the most flexible approach to reducing the data retrieval delay because of the short-stretch communication path between the cached data and consumers. Consequently, a cooperative caching-based scheme named Population-based Cache Policy (CSPC) was proposed by Chen et al. [27], using the Named Data Networking (NDN) in-network caching idea for vehicles to improve overall V2V network performance. A multi-tire-based cooperative caching strategy is proposed by Chen et al. [87] to enhance the data delivery services for connected vehicles. Based on the popular data items, the cooperative caching decision is performed. Popularity Prediction-Based Cooperative Cache Replacement (PPCCR) is proposed by Zhao et al. [88]. Based on the popularity prediction, the vehicles are rated to cache the data items. The replacement algorithm is used to predict the popular data items during a specific period of time.

In Figure 6, the cache-based V2V network is illustrated in which some of the vehicles are associated with caching capabilities to store the most frequently requested data items, and the connected vehicles can retrieve the required data items from these vehicles as shown in the given Figure 6.

Therefore, cooperative caching plays a significant role in enhancing the performance of overall ad hoc networks. Ad hoc networks include a number of performance metrics to evaluate the effectiveness of a network system. The important performance metrics are energy consumption, data retrieval latency, average access delay, cache hit ratio, packet delivery rate, average hop count, packet loss, byte hit ratio, stretch ratio, cache utilization, diversity ratio, data transmission efficiency, message complexity, and average traffic ratio and communication overhead. Based on these performance metrics, this section provides the cooperative-based caching contributions with caching models proposed. Table 2 illustrates the cooperative-based caching models and their basic contributions to enhancing the ad hoc network performance in terms of the above-mentioned performance metrics.

## 5. Summary and Discussion

MANET is a prospective technology that is very useful for the outdoors, aggregated meetings, and disaster or battlefield implementation. Nonetheless, MANETs have some issues in common with other wireless networks: high energy consumption, mobility, and inefficient route discovery affecting QoS. Cooperative caching provides a workable approach to these difficulties by enhancing communication facilities, saving energy and bandwidth, and shortening data access time. This approach makes it possible for nodes to pass on cached information, thus many nodal forms being able to access cached information to address their requirements. The ICP is developed by Claffy and Wessels, supports such functionality, and has led to the standardization of caching protocols into two categories: It can be divided into two categories based on the router: the router-based and the message-based. Router-based protocols are summary-cache and cache-digest, which help in the sharing of information among proxies and the distribution of user requests among cache arrays, while message-based are at the message level.

As the number of users of wireless networks increases, there is a need to increase their QoS and network connectivity without incurring the high cost of installation. Among such advancements is WMN technology, which provides network connectivity and coverage through multi-hop. WMNs offer the system flexible access opportunities and high-bandwidth communications. Yet they are shaped by several constraints, such as device portability and wireless communication resources, which remain scarce, and challenges that affect the efficiency of data access, threatening to hamper their advancement. Cooperative caching has therefore been developed as a solution to these challenges and involves the storage of popular data items in intermediate caching positions. The benefit of this approach is that desirable data becomes more readily accessible, making it less expensive and resource-consuming. Information stored at intermediate, requesting, or neighboring nodes can be transferred over multi-hop paths, which can reduce transmission costs and required resources.

This variability, along with the availability of a huge amount of data from smart sensors and wearable devices, has beautified the receptive function of WSNs in real-time applications. The potential applications of WSNs include disaster management, surveillance, health care, and environmental monitoring. Such networks comprise many sensor nodes required to collect, process, and transmit data to the base stations through single or multi-hop modes. Every sensor has a power supply module, a sensing element, a signal processing part, and a transmitting section. However, energy efficiency is still a big issue in WSNs with the constraints of the limited energy capacity of sensors, especially in adverse conditions. Energy problems have been tackled using different strategies, namely duty cycling, energy-aware MAC, low-power communication, and hardware layer protocols. However, energy efficiency remains an issue of considerable importance in WSNs. However, cooperative caching is known to present a rather compelling approach to increasing the energy efficiency of WSNs. When the transmitted data is cached at several nodes, the subsequent user requirements can be effectively fulfilled without requiring the same information back from the source, which again conserves energy and lessens the latency time. Data cached at a node can be retrieved back, including if the source node is in dormant or sleep mode, reducing energy consumption even more. This approach also reduces the congestion of the network and optimizes the general system by allowing quicker access to the cached nodes within the local networks. 

VANET holds high-speed ground vehicles that are connected to support communication in a tentatively self-organizing network through multi-hop data broadcast in the absence of fixed infrastructure. To gain accessibility and dynamic network connectivity, VANETs are therefore interfaced with wireless technologies, namely local & wide area networks. Such integration has vested VANETs as a prospective international communication network. While compared with cellular networks, VANETs facilitate a larger variety of heterogeneous services and applications and have stricter system QoS requirements for intelligent driving. The services include real-time traffic flow, parking space status, mobility of environment, and distances to nearby highways. Since the driving environment constantly evolves, timely information is crucial, and that constitutes a major problem for VANET. Another challenge is the management of the joint use of different kinds of resources for realizing the complicated QoS demands of mobile customers.

The automobile industry and wireless communication industries have lately directed their attention to enhancing VANETs using V2I and V2V, which operate on the IEEE 802.11p standard [89]. Such progress has been effective for road safety and traffic flow control with the help of the implementation of the ITS. For instance, data dissemination playing a very vital role in improving the performance of V2V technology has made the advancement of ITS attractive. It enables efficient road and traffic maintenance by transmitting key information about the location of an accident, for instance, and offers entertainment and news to the passengers. However, V2V systems demand highly reliable and high-rate data communications to enable enhanced transfer of enormous and rich multimedia messages such as photo images, musical notes, and video clips between the vehicles. This type of caching has come as the best solution in the distribution of V2V content through ITS and traffic systems for the best throughput. Caching mitigates the path stretch incurred to access specific data items since these data items can be redeemed for caching many varieties of equipment devices and vehicles. It allows for the provision of requests during high traffic volume by means of the cached data at low traffic moments, thus minimizing the reliance on the wireless base stations. With direct access to cache content, link and traffic overloading at base stations and in the backhaul network are greatly reduced. However, caching helps cut the time taken to load specific data since its communication is local to the consumer.

Emerging wireless network technologies, like MANETs, WMNs, and VANETs, have shown concern with issues such as limitations in resources, data access time, and higher communication bandwidth. Cooperative caching is now seen as a rather innovative and promising solution to increase the performance and optimization of such networks by storing requested data items in intermediate or edge nodes. This reduces dependence on central resources, fastens data access, and optimizes the network quality of service (QoS). MANETs is made cooperative caching useful and proficient in handling mobility and energy consumption issues and several nodes cached data and updated caching and a kind of hybrid cache management. This reduces overall end-to-end delays, packet losses, and energy consumption. Likewise, the WMNs use caching to store and forward messages locally, and by doing so, minimize not only the amount of transmission but also the resources consumed. Techniques such as CACC and CacheRescue are used to enhance the performance indicators, including cache-hit ratio, data access rate, and other network measures.

In the case of VANETs, caching improves the V2V messages since it is swift and reliable to disseminate information within the system. Through cache data items, reliance on wireless base stations is minimized, thus offloading the network for real-time requests on important and entertainment data. Techniques such as the CSPC and multi-tier cooperative caching enhance the delivery of content and traffic distribution based on the extent of the popular data items. Altogether, caching enhances these WLAN technologies in terms of reducing the time taken, the amount of bandwidth required, the flexibility of caching with dynamic networks, and the effectiveness of the data delivery. It is an essential component for meeting the concerns of current applications, maintaining fluent interaction, and improving the usability of interactions in various contexts.

## 6. Advantages of Cooperative-Based Caching

Cooperative-based caching is one of the most beneficial approaches to improving the overall wireless data dissemination performance by providing cache storage to the transmitted data items. As a result, the data items can be cached at intermediate locations to accomplish future requests for the data [90]. It delivers several benefits to wireless-based Internet technologies, such as flexible data retrieval, lower bandwidth utilization, reduction of server load, small stretch path, minimization of energy consumption, and minimization of cost. Moreover, it also reduces the burden on front-haul and back-haul links and significantly decreases delay. Therefore, the content-aware and delay-sensitive applications are executed near the desired users [91,92].

### 6.1. Server Load and Bandwidth Reduction

The faster attainment of mobile devices has brought the production of several kinds of data traffic, such as Augmented Reality (AR) and Virtual Reality (VR). According to the VNI forecast, the AR and VR traffic will increase 7 and 11 times, respectively, in the next few years. These types of traffic usually need extensive resources to disseminate content between distinct locations and require communication services that impose stress on network links and produce critical problems related to spectrum scarcity due to high bandwidth requirements [93]. Therefore, the high usage of bandwidth due to spectrum scarcity becomes a critical problem for network operators. Recently, multicast mechanisms have been implemented through cooperative caching services in wireless-based networks. Cooperative caching is the most promising approach to handling wireless spectrum and bandwidth-related problems efficiently. Usually, modern data traffic requires heavy computation responsibilities, such as online gaming and AR/VR traffic delivery. The cache-able network environments offer a favorable paradigm to enhance the user-perceived QoS by processing data items with less complexity and bringing them near the end-user at edge network nodes (edge computing). The cache-able nodes are used to minimize the peak-time data traffic by prefetching the popular data items and caching them close to the users during off-peak time [49]. As a result, the backhaul capacity and user-perceived QoS improve in wireless-based networks.

### 6.2. Cost/Economic Implications

In this subsection, we explore how cooperative-based caching is deployed and used in different types of network players, such as the Content Provider (CP) and Internet service providers (ISPs). Caching accelerates the distribution of contents to each network node equipped with cache storage. In several studies, it has been observed that caching helps ISPs to reduce their inter-domain traffic cost by caching their required data items at the local cache [94]. For example, the transit cost will be excluded if the item of data is needed in a cache in the ISP’s domain near the end user. The ISPs can achieve high-quality data delivery services through the implementation of caching because they have the detailed status of the network infrastructure, such as link status information. Cooperative caching is considered a means to achieve advantages not only from the perspective of the consumer (e.g., reduced content retrieval latency and short stretch) but also in terms of economic profits. In a business model, the CP uses several resources and charges their consumers for discriminative data items [95]. Conversely, if ISPs have their caching infrastructure, they can provide data items to CPs. Therefore, the ISPs can charge for the data caching services. The reason is that all the ISPs want to profit, and they do not want to allow their cache resources to be used by others without any charge. The CPs want to get their desired data items without delay and at a minimum cost. In this scenario, caching provides several benefits (short stretch and low delay) and saves the high utilization of resources and energy by caching the required data items near the CPs. Several studies have recently been proposed to explore the economic implications of utilizing caching. These studies observed that the network nodes equipped with caching would obtain more incentives by delivering data items through caching services. Therefore, to increase the incentive reward, caching provides several cooperative-based cache placement strategies. The ISPs can merge their appropriate caching strategy according to their domains and offer better data caching services to minimize retrieval time, power consumption, resource consumption, and stretch [96].

### 6.3. Path Stretch and Delay Reduction

Currently, a considerable number of applications are running over the wireless-based Internet that connects heterogeneous devices through communications services. However, wireless-based networks need to optimize the communication services to enhance data retrieval that requires a short stretch path with minimum delay. Therefore, a cache-able network environment can solve these critical issues with high efficiency and flexibility. In the early decades, the researchers were focused on providing efficient research sources for data dissemination. However, the amount of communicating data has to turn into a gigantic amount because of the end-to-end communication network infrastructure. Therefore, caching is the most beneficial approach to implement at the network edges. The cache can be deployed at different types of BSs, such as SBS, MBS, and PBS, to minimize the path stretch and data retrieval delay [97]. In a simple wireless-based network, the data needs to be downloaded from the remote providers, and instead of caching the popular data items at an intermediate location, it only delivers them to the requesters. On the other hand, in a cache-based wireless network, the transmitted data items can be cached at different places (at base stations and user devices in the D2D network). Also, in wireless-based heterogeneous networks, the SBSs are distributed densely. Thus, the cache-able SBS is the right choice since the SBSs are deployed closer to the end-users, which delivers a high data rate. As a result, network performance can be increased in terms of stretch and delay [98].

### 6.4. Energy and Power Saving

The Small Cells (SCs) are offering many approaches to overcome the exponentially increasing wireless-based data dissemination. However, energy efficiency is continuously decreasing in ad hoc networks because of the additional distribution of SBs. Therefore, energy and power consumption have appeared as the most crucial issues in ad hoc-based network infrastructure [99]. The purpose of efficient power and energy consumption is to provide benefits for network operators and to improve QoS by extending the battery life of the mobile and terminal nodes, which can be achieved by the implementation of enhanced network architecture with less signaling overhead. Recently, incredible efforts have been made for the advancement of wireless-based communication. However, the continuously rising need for energy and power in ad hoc-based networks has become crucial for efficient data dissemination. At the same time, there is a shift from traditional-based wireless Internet to caching-based network infrastructure that can reduce energy and power consumption. Therefore, a cache-able wireless network is a pivotal approach to significantly improving performance in terms of minimizing the strength and power consumption of wireless-based communication networks. In a sensor network, the lifetime of a battery can be extended by reducing the amount of disseminating data, and this can be done by caching the most popular data items close to the end users [100]. Likewise, in D2D data dissemination, a mobile device in proximity can be integrated with cache storage to establish a short stretch in data downloading to save energy. Besides, the end-users can also download the popular data items from multiple caching sources, even a node in sleeping mode and it does not require sending the requests to the main source (macro-cell base stations), keeping the energy usage at its minimum level.

## 7. Lessons Learned and Key Insights

In the present survey, it is clear that wireless-based communication technologies are suffering from a number of critical challenges, such as query delay, power consumption, resource utilization, bandwidth, offloading, backhauling, link capacity, and energy efficiency. Cooperative-based caching is considered the most suitable approach to deal with these vital challenges that can handle these challenges with high efficiency. It has a lot of advantages over traditional wireless-based network environments, and most researchers are exploring caching capabilities to enhance the wireless infrastructure. Caching is one of the most flexible approaches to improving the performance of wireless-based networks. To this end, a large number of cache deployment schemes were developed to implement in different types of wireless-based networks. Most of these solutions may be a shortfall for future Internet requirements due to the high volume of data and heterogeneity of devices. The incredible growth of today’s Internet traffic and frequent transmission of similar content generate several problems in which bandwidth utilization, high usage of resources and power consumption are most important. These problems are hard to resolve by using the traditional Internet paradigm. Therefore, it needs to change the design of Internet architecture because the consumer is only interested in retrieving their preferred content rather than the physical locations of hosts. Moreover, the current Internet offers a peer-to-peer communications system that is insufficient to handle the growing issues to arrange effective content dissemination. In this situation, in-network cache storage can be deployed to enhance the content dissemination process by caching the disseminated contents at intermediate locations for subsequent content routing.

Ad hoc-based cooperative caching and newer-age technologies, such as 6G, IoT, and edge computing, hold great opportunities for improving present networks in terms of efficiency, scalability, and adaptability. These integrations build on the capabilities of each technology to overcome the issues with traditional caching while anticipating and providing solutions for future communication and data management issues.

The 6G networks are characterized by very high data rates, even higher network availability, an increased number of connected devices, and unique artificial intelligence capabilities. Such developments will be of immense help in cooperative caching in mobile ad hoc networks. For example, in 6G, the AI of 6G can be applied to predict demands from the user and preload the content at ad hoc nodes, resulting in better cache hit ratios and lower latency. Finally, 6G’s edge architecture also matches the decentralized architecture of ad hoc networks to ensure that content can be stored closer to the end users. In addition, the improvements in both spectrum efficiency and resource allocation in 6G enable cooperative caching to be changed based on the needs of a particular network, increasing scalability and decreasing crowding.

Moreover, smart devices have flooded the market and produced enormous volumes of data; therefore, there is a need to store the data and make it easily available all the time. These challenges can be solved using ad hoc-based cooperative caching that will allow IoT devices in the same region to download data from close devices instead of depending on the central server to provide information, which will take a long time. This is particularly beneficial to battery-driven IoT instruments since it helps to conserve power by not continuously requesting data from distant servers. In addition, cooperative caching can help achieve scalability by dividing the tasks of caching and the access to multi-IoT device data. For some specific IoT applications that are considered sensitive applications, like healthcare IoT or industrial IoT, caching mechanisms can be used to cache important data to facilitate access and achieve high availability.

Edge computing is considered a processing and data storage technology that occurs at the farthest point from the data source, device, or the cloud to reduce both edge distance, hence time, and consumed bandwidth. Thus, cooperative caching can complement edge computing in such a way as to employ edge nodes as caching points for ad hoc networks, which should store frequent accesses to reduce the certify loads. The computational capabilities of edge nodes also include real-time processing capabilities, which make it possible to make caching policies adapt to dynamic network conditions such as content popularities or nodal mobility. However, edge computing, which processes data in the vicinity, provides more security and, therefore, along with ad hoc cooperative caching, guarantees secure, efficient, and localized data access.

However, including the idea of ad hoc cooperative caching in 6G, IoT, and edge computing has its drawbacks. Compatibility with other systems is also a major issue because they have to integrate into our current systems to provide the required interface. Security and privacy are the other main shortcomings in IoT and subsequently, 6G applications, as they involve the conveyance of sensitive information. Furthermore, AI-based caching solutions for IoT impose a computational cost that should be considered in correlation to the energy constraints of IoT devices. The dynamic nature of ad hoc networks and edge environments demands real-time adaptation to the environment, which in turn requires the innovation of complex caching algorithms. Solving these issues enables ad hoc cooperative caching to be integrated with 6G, IoT, and edge computing as breakthroughs toward the intelligent automation of key innovation sectors such as autonomous vehicles, smart cities, and the Industrial Revolution 4.0.

Information-Centric Networking (ICN) provides a large number of network-based caching approaches for the diverse kinds of wired and wireless-based networks. ICN has been providing various caching placement strategies to cache the appropriate data items at suitable places. Moreover, ICN provides replacement policies to avoid the unnecessary usage of cache to improve the performance of a network environment. In ICN, users can send their interests in certain content through multicast, broadcast, and anycast. ICN uses three kinds of data structures: Content Store (CS), Pending Interest Table (PIT), and Forwarding Information Base (FIB). The CS is responsible for storing the content, the PIT keeps records of the pending interests, and the FIB is responsible for sending the incoming user’s request to the appropriate source. On the other hand, if the request packet does match with any of the existing entries at a caching router, the interest is forwarded toward the suitable content container [45].

In the last decade, several ICN-based caching strategies were developed in which cooperative caching achieved more interest from researchers. For instance, an age-based cooperative caching system was proposed to minimize the network delay and publisher load. This strategy depends on the lightweight collaboration mechanisms to cache the disseminated contents at the network edges. Consequently, the cache of the intermediate nodes is used efficiently. By caching the content at network edges, the overall network delay and publisher load become reduced to fulfill the subsequent end-user requirements. Similarly, the caching facilities are also broadly used in ICN-based wireless networks. For example, popularity-based caching strategies and dynamic probability-based caching strategies were developed to enhance the ICN-based vehicular ad hoc network, such as the ICN-based Cooperative Caching (ICoC) strategy and Storage Planning and Replica Assignment (SPRA) [23]. Indeed, the content is cached within vehicle storage that is used to fulfill the subsequent future requirements of the neighboring vehicles. Moreover, the data access rate is improved, and the load on the content provider is reduced by caching content using cooperative-based caching strategies.

Therefore, the overall network performance is improved in terms of content availability, average time delay, communication overhead, network throughput, and content retrieval latency through cooperatively caching the popular contents at intermediate nodes (vehicles). In short, we expect this study will open opportunities for a better understanding of current cooperative caching solutions and provide insights into new architectural designs. This study may also be useful for the deployment of emerging technologies such as fog computing, edge computing, software-defined networking, and the Internet of Everything (IoE). For the end-users, these caching solutions are the most beneficial to lessen the response latency, shorten the stretch path, reduce the link load, and improve the data availability with less delay.

## 8. Conclusions

This paper offers a systematic literature review on cooperative caching in ad hoc incorporated wireless communication systems, such as MANETs, WMNs, VANETs, WSNs, and IoT technology networks. The review further looks at the potential of caching to improve the network’s performance based on scalability and energy consumption despite characteristics like redundancy, mobility, and latency. This work provides a classification of existing caching approaches and insight into their ability to be practically applied in various network scenarios. The survey also amplifies the prospects of cooperative caching in conjunction with future technologies, including 6G, IoT, and edge computing for better data delivery, less delay, and better use of resources. There is potential for these integrations to solve traditional network issues and allow better-designed and more resilient communication networks. However, the survey also reveals some research limitations where security aspects, energy efficiency and trade-offs, and system interoperability in dynamic networks have been poorly addressed. Based on the ICN integrated solution with other networking technologies, we concluded that ICN caching is more feasible for other networking fields and can deliver several benefits in which short response latency, low bandwidth, low congestion, and low power and resource consumption can be provided. We believe this survey will stimulate the research community and pave the way toward the empowerment of emerging networks to enable efficient data dissemination.

## Figures and Tables

**Figure 1 sensors-25-01258-f001:**
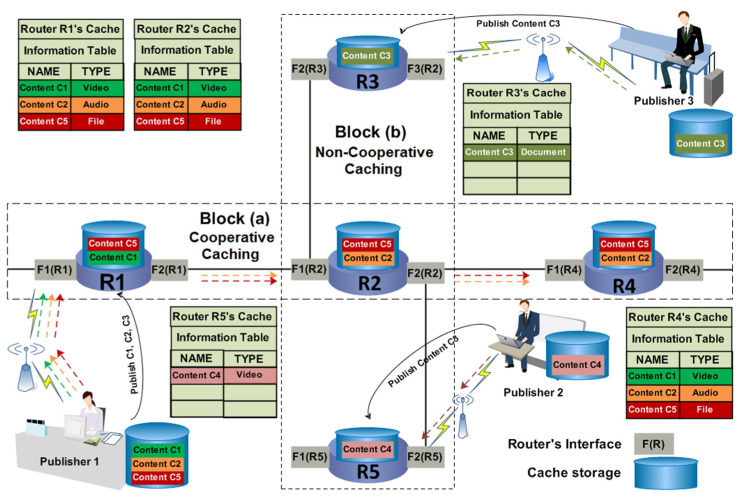
Cooperative versus non-cooperative caching.

**Figure 2 sensors-25-01258-f002:**
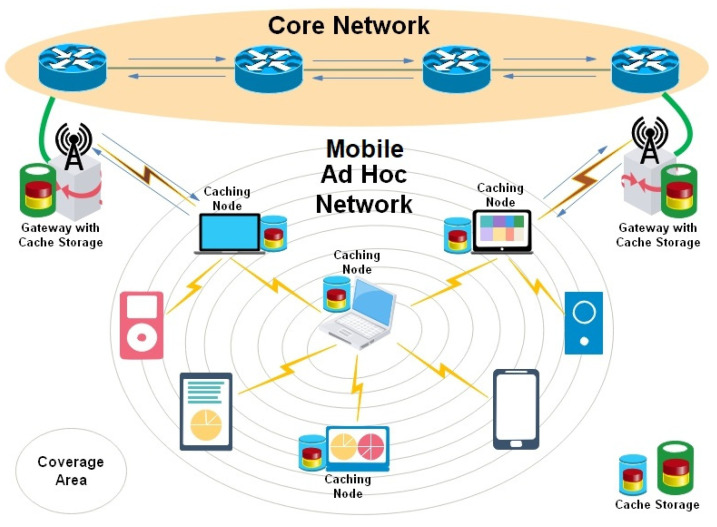
Cooperative Caching in Mobile ad hoc Network.

**Figure 3 sensors-25-01258-f003:**
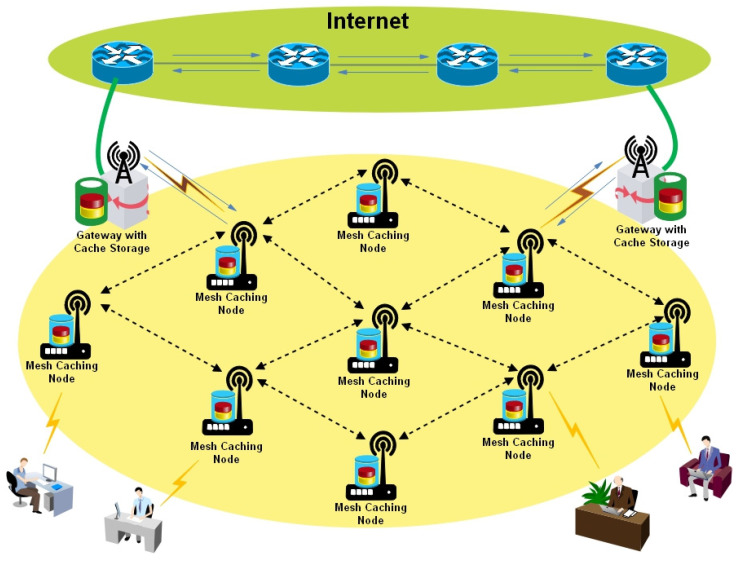
Cooperative Caching in Wireless Mesh Network.

**Figure 4 sensors-25-01258-f004:**
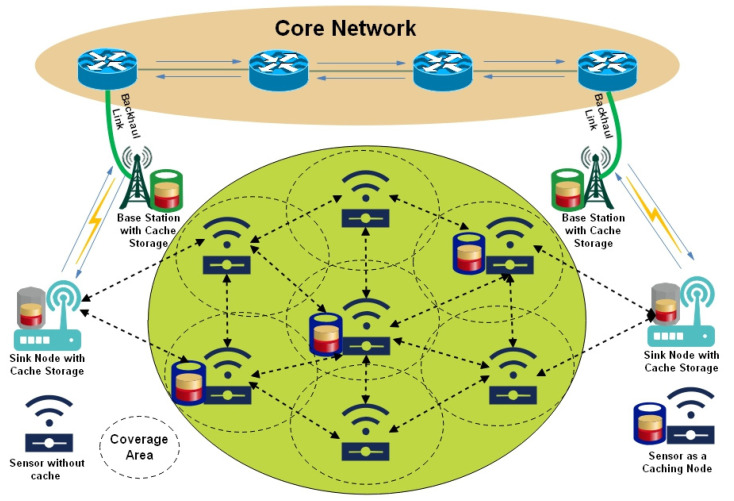
Cooperative Caching in Wireless Sensor Network.

**Figure 5 sensors-25-01258-f005:**
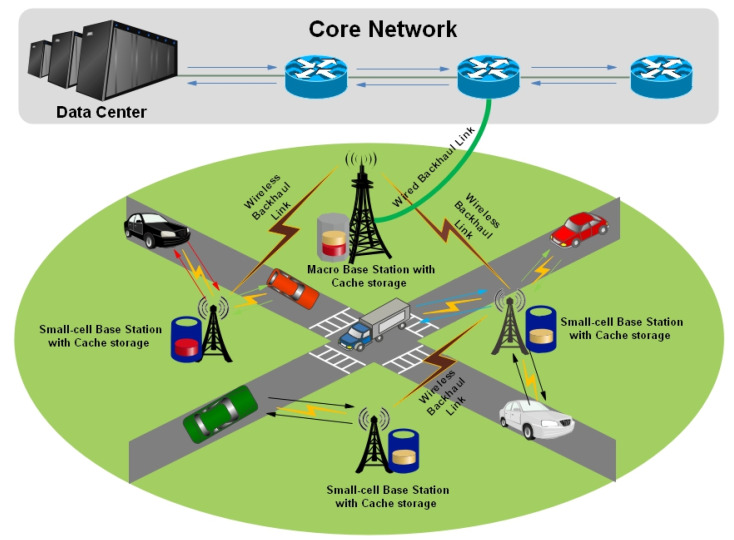
Cooperative Caching in Vehicular Ad hoc Network.

**Figure 6 sensors-25-01258-f006:**
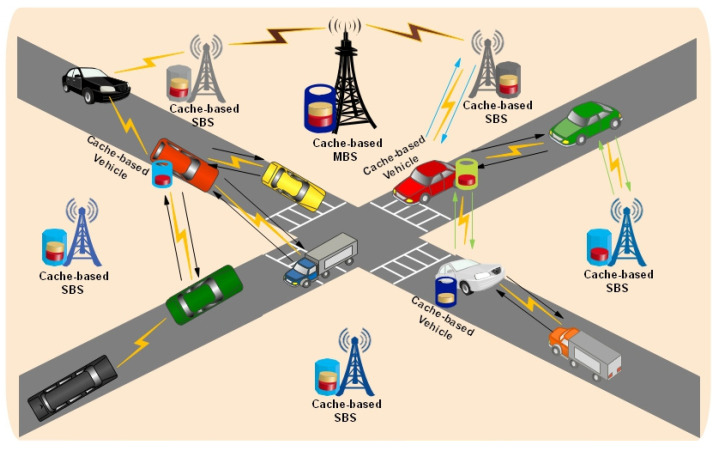
Cooperative Caching in Vehicle to Vehicle Wireless Network.

**Table 1 sensors-25-01258-t001:** Summary of existing surveys.

Year	References	Primary Focus of Existing Surveys
2019	Jain et al. [25]	In this the survey, the mobile-based ad hoc network is thoroughly explained and cooperative caching is suggested as the most significant method to enhance the performance of MANET.
2020	Lin et al. [26]	In this the survey, a cooperative-based mobile network caching with coordinated multipoint joint transmission is proposed to enhance future cellular networks.
2020	Chen et al. [27]	Both a survey on vehicular named data networking data caching and replacement strategies are critically explained.
2020	Mohajer et al. [28]	This study outlines cache-based communication, beamforming, interface mitigation, and resource allocation.
2021	Ahmed et al. [29]	In this study, caching types and techniques to improve 5G network performance are described.
2021	Shuja et al. [30]	This study investigates machine learning-aware caching schemes at network edges. Caching-based data replacements policies are also described.
2021	Naeem et al. [31]	This study provides a comprehensive survey on named data networking-based caching strategies developed to enhance Internet of Things-based environments. Also, a comparative analysis of strategies is discussed.
2022	Yang et al. [32]	This study focuses on the NOMA-based FRAN architecture using cooperative caching and cache-based mobile edge computing; artificial intelligent techniques are used to solve the NOMA-based FRAN issues.
2022	Wu et al. [33]	In this study, reinforcement learning-based mobile edge caching techniques to optimize the 6G cellular network’s performance are discussed.
2022	Bepari et al. [34]	This survey investigates vehicles’ mobility in order to design cooperative caching using machines learning concepts.
2022	Al-Ward et al. [35]	This study explains caching based on non-orthogonal multiple-access technologies for the 6G cellular wireless network.
2022	Duong et al. [36]	This study provides knowledge about Information-Centric Internet of Things (IC-IoT)-aware caching mechanisms and data freshness values.
2023	Van et al. [23]	This study explores in-network caching strategies in Information-Centric Networks (ICNs), including MANET, VANET, IoT, and WSNs, discussing trends, issues, and future research.
2024	Ahmed et al. [24]	This study explores caching techniques in wireless ad hoc networks, addressing data duplication, mobility, and location-dependency issues, as well as highlighting NDNs’ advantages, such as scalability, security, and privacy.
2024	Obaid et al. [19]	This systematic review examines caching patterns in ad hoc networks, focusing on performance, size, vitality usage, and fluid topology adaptability.

**Table 2 sensors-25-01258-t002:** Cooperative Based Caching in Ad hoc networks.

Year	Reference	Technology	Simulator	Aim/Goal
2022	Satyanarayana et al. [60]	MANET	NS2	This paper proposed a mechanism which improves packet delivery ratio, energy consumption, end-to-end delay, and packet loss in VANETs by up to 35–45%
2021		MANET	NS2	HCM improve the network performance in terms of query delay, message complexity, and hit ratio.
2020	Sheeba et al. [61]	MANET	NS2	It enhance access latency, cache hit ratio, communication overhead, and average traffic ratio in MANET.
2021	Kuila et al. [68]	WMN	NS3	ACRM improve the WMN performance in terms of network stability and data transmission efficiency.
2014	Xu et al. [65]	WMN	IEEE 802.11 DSDV	The results of this strategy show that it can save message cost by 50–70% in a WMN.
2022	Chen-Nakesavula et al. [72]	WSN	NS2	It improves the performance of WNS in terms of PDR of nearly 50–60%.
2022	Dinh et al. [73]	WSN	NS3	It improve the cache hit ratio and overall WSN network performance
2022	Yang et al. [74]	WSN	COOJA	TCM caches popular data at the sink and within the WSN that improve hit ratio by 28.4%, stretch ratio, and average latency by 1.13 s.
2022	Gupta et al. [80]	VANET	ONE	It enhances the average access delay, cache hit ratio, average hop count in VANET.
2021	Amadeo et al. [81]	VANET	ndnSIM	It enhances the caching performance in terms of latency, hit ratio, hop count, cache utilization, and diversity.
2021	Chen et al. [82]	VANET	MATLAB	It cache the content in the chuck by chunk to improve the hit ratio, average access delay, and average hop count
2020	Zhao et al. [83]	VANET	ndnSIM	improve the content retrieval delay, success ratio, and resource utilization in VANET
2020	Chen et al. [27]	V2V	ndnSIM	It improves the V2V performance in terms of data transmission efficiency and resource utilization.
2020	Chen et al. [87]	V2V	RSU	It reduces the services cost and transmission delay, coverage, reliability, and robustness.
2018	Zhao et al. [88].	V2V	OMNet++	PPCCR reduces the average delay, improve the cache hit ratio, and enhance the cache hit distance.

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
