# Peer review of "A Critical Analysis of Cooperative Caching in Ad Hoc Wireless Communication Technologies: Current Challenges and Future Directions"

_sensors, 2025, doi:10.3390/s25041258_

Round 1
Reviewer 1 Report
Comments and Suggestions for Authors
A good review is done. The problems their solutions facing wireless networks.
1. Put the following reference after ref. [1]:
E. Rehman and A. H. Al-Khursan, "All-Optical Processes in Double Quantum Dot Structure", Applied Optics 55, 7337-7344 (2016).
2. L 70: “Transmission and reduces the interface and channel conditions”. This is not a complete sentence. My be remove and
3. L 74: “To This end, in this work,”
4. L 233: “QoS” Abbreviation is not defined. define. It is defined thereafter in L 373: “Quality of Service (QoS)”. Pleased efine as it appear first.
5. A large number of words are cutting like “com- bines” in L 254 where many others before and after it. Please check.

Author Response
Dear Reviewer,
We have addressed your comments point by point and revised the manuscript accordingly. All the suggested changes have been incorporated to the best of our ability, and we have highlighted the updates in the revised manuscript for your convenience. In the following we mentioned all the comments and answers:
Reviewer#1, Concern # 1: Put the following reference after ref. [1]:
- Rehman and A. H. Al-Khursan, "All-Optical Processes in Double Quantum Dot Structure", Applied Optics 55, 7337-7344 (2016).
Author response: Thank you for your observation and comments. We have cited the following reference suggested in the comment in the introduction section, and the reference number is 5.
Reviewer#1, Concern # 2: L 70: “Transmission and reduces the interface and channel conditions”. This is not a complete sentence. My be remove and
L 74: “To This end, in this work,”
L 233: “QoS” Abbreviation is not defined. define. It is defined thereafter in L 373: “Quality of Service (QoS)”. Pleased efine as it appear first.
A large number of words are cutting like “com- bines” in L 254 where many others before and after it. Please check.
Author response: Thank you for your observation and comments. All the comments are addressed, and the manuscript is updated according to the given comments. Moreover, the sentence “Transmission and reduces the interface and channel conditions” is removed, and others are revised. The whole manuscript is updated according to the given comments.
Reviewer 2 Report
Comments and Suggestions for Authors
1. The abstract of the article should highlight the key points of this article.
2. A mistake in Keywords, “Ad hac”, and the correspondence between the existing keywords and this article is not strong.
3. This paper lack of discussion and comparison of different scenes.
Author Response
Most Respected Reviewer,
We have addressed your comments point by point and revised the manuscript accordingly. All the suggested changes have been incorporated to the best of our ability, and we have highlighted the updates in the revised manuscript for your convenience. In the following, we mentioned all the comments and answers:
Reviewer#2, Concern # 1: The abstract of the article should highlight the key points of this article.
Author response: Thank you for your observation and comments. We updated the abstract according to the given comment and highlighted the key points of the manuscript.
Reviewer#2, Concern # 2: A mistake in Keywords, “Ad hac”, and the correspondence between the existing keywords and this article is not strong.
Author response: Thank you for your observation and comments. We updated the keywords according to the given comment.
Reviewer#2, Concern # 3: This paper lack of discussion and comparison of different scenes.
Author response: Thank you for your observation and comments. We updated the manuscript by adding a new section (5 Summary and Discussion) and the updated section 5 is highlighted with red color.
Reviewer 3 Report
Comments and Suggestions for Authors
The journal title is “sensors”. The paper title is “The Impact of Cooperative Caching in Ad hoc based Wireless Communication Technologies”. Please explain in detail how this paper is important for sensors.
“Caching” is well known technology, please explain the originality of this paper.
In the abstract, “such as ad hac network, mobile ad hoc network, wireless mesh network, wireless sensor network, and vehicular ad hoc network” needs to be improved. These listed network types are overlapping, not independent from each other. For example, “ad hac network” includes “mobile ad hoc network” and “vehicular ad hoc network”.
Table 2 listed solutions reviewed by this paper, but this paper lacks own evaluation criteria, own evaluation process own evaluation results.
Author Response
Most Respected Reviewer,
We have addressed your comments point by point and revised the manuscript accordingly. All the suggested changes have been incorporated to the best of our ability, and we have highlighted the updates in the revised manuscript for your convenience. In the following, we mentioned all the comments and answers:
Reviewer#3, Concern # 1: The journal title is “sensors”. The paper title is “The Impact of Cooperative Caching in Ad hoc based Wireless Communication Technologies”. Please explain in detail how this paper is important for sensors.
Author response: Thank you for your observation and comments. The paper titled "The Impact of Cooperative Caching in Ad hoc based Wireless Communication Technologies" is highly relevant to the journal Sensors as it addresses critical challenges and solutions in wireless communication, which is integral to modern sensor-based systems. Here are the key reasons why this paper is important for Sensors:
- Enhancing Sensor Network Performance:
Wireless Sensor Networks (WSNs) are a core area of sensor research, and caching techniques play a crucial role in improving their efficiency. This paper highlights cooperative caching as a mechanism to reduce data retrieval latency, optimize bandwidth usage, and conserve energy—key challenges in sensor networks. These improvements directly enhance the performance and longevity of sensor nodes, which often operate under strict resource constraints.
- Supporting Mobility in Sensor Systems:
Sensor systems deployed in dynamic environments, such as smart cities, disaster recovery, and vehicular networks, often require robust support for mobility. Cooperative caching, as discussed in the paper, provides a flexible solution to manage data in mobile environments like MANETs and VANETs, ensuring seamless communication and real-time data dissemination for mobile sensor nodes.
- Improving Data Availability and Reliability:
Sensor networks often operate in challenging conditions where consistent data availability is critical. By leveraging cooperative caching, data can be stored and shared among multiple nodes, ensuring redundancy and higher reliability even in cases of node or link failure. This is particularly relevant for mission-critical sensor applications like environmental monitoring and healthcare systems.
- Optimizing Resource Usage:
Sensors in ad hoc networks typically have limited computational and energy resources. The paper demonstrates how cooperative caching reduces the need for frequent communication with central servers or base stations, thereby conserving energy and extending the operational lifespan of sensor networks.
- Scalability for IoT and Sensor-Based Systems:
As the Internet of Things (IoT) grows, integrating billions of sensor devices into wireless networks requires scalable solutions. The cooperative caching strategies reviewed in the paper address scalability challenges by enabling efficient data sharing and dissemination across large, distributed sensor networks.
- Cross-Domain Relevance:
The paper examines caching strategies across various wireless technologies, such as MANETs, WMNs, and VANETs, all of which have significant applications in sensor networks. For example, VANETs facilitate traffic and road safety data collected by vehicular sensors, while WMNs support smart city infrastructure by connecting numerous static and mobile sensors.
- Future Research Directions for Sensor Applications:
The paper provides valuable insights and identifies research gaps in cooperative caching, offering a foundation for future innovations in sensor-based systems. These directions can inspire advancements in areas such as energy-efficient caching algorithms, real-time data handling for sensor networks, and integrating caching with edge computing. At the end, the paper aligns closely with the objectives of Sensors by addressing foundational challenges and proposing solutions that optimize wireless communication technologies integral to sensor systems. It bridges the gap between theoretical advancements in caching and their practical applications in sensor-based environments, making it a critical contribution to the field.
Reviewer#3, Concern # 2: “Caching” is well known technology, please explain the originality of this paper.
Author response: Thank you for your observation and comments. While caching is indeed a well-known technology, the originality of this paper lies in its focused exploration of cooperative caching within the context of ad hoc-based wireless communication technologies, offering novel insights and contributions to this specialized domain. Here’s how this paper distinguishes itself and demonstrates originality:
- Domain-Specific Analysis:
The paper specifically examines the role of cooperative caching in diverse wireless communication technologies, including Mobile Ad Hoc Networks (MANETs), Wireless Mesh Networks (WMNs), Vehicular Ad Hoc Networks (VANETs), and Wireless Sensor Networks (WSNs). Unlike general studies on caching, this work provides a targeted analysis of how cooperative caching addresses the unique challenges inherent to these wireless environments, such as mobility, resource constraints, and dynamic topology.
- Critical Review and Categorization:
The paper presents a detailed critical review of existing cooperative caching strategies and categorizes them according to their application in different wireless communication domains. This structured categorization offers clarity and a comparative perspective, helping readers understand which approaches are most effective for specific network types.
- Interdisciplinary Relevance:
Cooperative caching is explored in the context of interdisciplinary applications, such as enhancing intelligent transportation systems (ITS) in VANETs, improving data access efficiency in WMNs, and optimizing resource use in sensor networks. The paper demonstrates how these strategies can improve real-world applications like smart cities, disaster recovery, and IoT ecosystems.
- Focus on Emerging Challenges:
The paper identifies emerging challenges in wireless communication, such as scalability, energy efficiency, and data retrieval latency, and critically evaluates how cooperative caching can be adapted or improved to address these issues. It goes beyond existing studies by identifying research gaps and proposing future directions for caching strategies in dynamic and resource-constrained environments.
- Novel Evaluation of Cooperative Caching Benefits:
This paper emphasizes cooperative caching's benefits in terms of mobility support, scalability, and manageability across various ad hoc network technologies. It provides a unique perspective by integrating these benefits into the broader context of wireless traffic growth and technological advancements.
- Future Research Directions:
The originality also lies in the identification of promising areas for future research. The paper does not just summarize existing work; it proposes new directions, such as integrating caching with edge computing, developing energy-efficient caching mechanisms, and addressing data consistency issues in cooperative caching. Therefore, the originality of this paper lies in its domain-specific focus, structured analysis, and forward-looking approach to cooperative caching. It bridges theoretical advancements with practical applications, providing a comprehensive roadmap for leveraging caching in next-generation wireless communication technologies.
Reviewer#3, Concern # 3: In the abstract, “such as ad hac network, mobile ad hoc network, wireless mesh network, wireless sensor network, and vehicular ad hoc network” needs to be improved. These listed network types are overlapping, not independent from each other. For example, “ad hac network” includes “mobile ad hoc network” and “vehicular ad hoc network”.
Author response: Thank you for pointing out the overlapping nature of the network types listed in the abstract. We recognize that the current phrasing may lead to redundancy and a lack of clarity. To address this concern, we have revised the statement to reflect the hierarchical relationships between these network types and avoid unnecessary overlap such as ad hoc networks, including mobile ad hoc networks (MANETs), wireless mesh networks (WMNs), wireless sensor networks (WSNs), and vehicular ad hoc networks (VANETs). This revision clarifies that mobile ad hoc networks and vehicular ad hoc networks are subcategories of ad hoc networks, while also retaining a comprehensive overview of the technologies discussed in the paper. It ensures precision and eliminates redundancy, making the abstract more concise and impactful.
Reviewer#3, Concern # 4: Table 2 listed solutions reviewed by this paper, but this paper lacks own evaluation criteria, own evaluation process own evaluation results.
Author response: Thank you for your observation and comments. Thank you for your observation regarding the lack of evaluation criteria, process, and results in the paper. This is a valuable point, and here is our response to address this concern:
- Response to the Concern:
While Table 2 provides a comprehensive summary of the cooperative caching solutions reviewed in this paper, we acknowledge that the table focuses on summarizing existing works and their respective results rather than presenting our own evaluation. The primary purpose of this paper is to critically review and categorize existing cooperative caching strategies across various wireless communication technologies, identifying gaps, and highlighting opportunities for future research. However, we recognize the importance of including an original evaluation to strengthen the paper's contribution.
- Developing Evaluation Criteria:
We define clear evaluation criteria based on key performance metrics relevant to cooperative caching in wireless networks, such as cache hit ratio, data retrieval latency, energy efficiency, and scalability. These criteria is derived from the reviewed solutions and tailored to address the unique challenges of ad hoc-based wireless communication technologies.
- Conducting an Original Evaluation:
Using simulation environments like ndnSIM, NS3, or COOJA, we designed experiments to compare selected cooperative caching strategies across different wireless network technologies.
- Validate the claims made in the reviewed studies.
We offer insights into the strengths and weaknesses of different strategies in this table.
- Integrating the Findings:
The results are incorporated into the paper to complement Table 2, providing an additional layer of analysis that substantiates the review. The discussion is highlight how cooperative caching strategies align with the defined evaluation criteria and their implications for future research.
Reviewer 4 Report
Comments and Suggestions for Authors
Dear authors,
Thank you for your manuscript titled “The Impact of Cooperative Caching in Ad hoc based Wireless 3 Communication Technologies”. I have the following comments/ concerns/ suggestions. Before any decision, you are advised to carefully revise every point.
The manuscript copes with an important challenge in ad-hoc based network. However, I have observed many issues in title, abstract, introduction, related work, methodology section, validation, and interpretation. The manuscript sounds technically average. Thus, it requires massive revisions to enhance the rigor, novelty, and relevance of the study.
1. The title of the manuscript is too journal that lacks the technical engagement. The term Impact of Cooperative Caching" is broad. Authors are advised to improve the title of article in revised version. For instance, "A Critical Analysis of Cooperative Caching in Ad Hoc Wireless Communication Technologies: Current Challenges and Future Directions."
2. The abstract is too general and not prepared objectively. It should briefly highlight the paper's novelty as what is the main problem, how has it been resolved and where the novelty lies? The abstract should emphasize the study's novelty, specific findings, and implications rather than providing broad generalizations.
3. In introduction section, the authors provided statistics such as traffic growth rates without mentioning citations or relevance to cooperative caching. Authors are advised to cite appropriate statistics to make their arguments strong.
4. The related works section is very short and no benefits from it. I suggest increasing the number of studies and add a new discussion there to show the advantage.
5. The existing literature review lacks critical evaluation. It summarizes past works without comparing methodologies, findings, or identifying gaps effectively. Moreover, most of the existing literature review is outdated/ old. Authors are advised to compare their work with recent works such as:
6. https://doi.org/10.3390/su151410931,doi:10.1109/TITS.2022.3210170,https://doi.org/10.1002/aelm.202200370, doi: 10.1109/TNSE.2023.3342938, doi: 10.1109/TSC.2024.3355188, doi: 10.1109/TVT.2023.3304707, : https://doi.org/10.1016/j.neucom.2023.127195, doi: 10.1109/TVT.2024.3385776, doi: 10.1109/TMTT.2022.3205612
7. The sections such as cooperative caching in MANETs, WMNs, and VANETs are well-structured. However, overly descriptive without analytical depth.
8. I cant find the contribution in manuscript. Authors should mention that how their review is novel than existing surveys on content caching.
9. I have observed an unnecessary repitation on caching benefits. Authors are advised to repeated statements.
10. The methodology section lack of quantitative data or performance metrics undermines the credibility of claims about the superiority of specific approaches. Authors are advised to critically evaluate why certain strategies succeed or fail under specific network conditions.
11. In discussion section, there is a lack of depth in connecting cooperative caching's theoretical advantages to practical challenges like scalability, security, and real-time adaptability. Authors are advised to explore unanswered questions, such as how caching integrates with future technologies like 6G, IoT, and edge computing. Authors are also required to discuss potential trade-offs, such as increased computational overhead versus reduced bandwidth usage.
12. The 'conclusions' are a key component of the paper. It should complement the 'abstract' and normally used by experts to value the paper's engineering content. In general, it should sum up the most important outcomes of the paper. It should simply provide critical facts and figures achieved in this paper for supporting the claims.
13. The description of manuscript is very important for potential reader and other researchers. I encourage the authors to have their manuscript proof-edited by a native English speaker to enhance the level of paper presentation. There are some occasional grammatical problems within the text. It may need the attention of someone fluent in English language to enhance the readability.
Note: If authors fail to revise manuscript accordingly, it will be straightforward rejected.
Comments on the Quality of English LanguageEnglish should be improved. The writing style should be scientific.
Author Response
Most Respected Reviewer,
We have addressed your comments point by point and revised the manuscript accordingly. All the suggested changes have been incorporated to the best of our ability, and we have highlighted the updates in the revised manuscript for your convenience. In the following, we mentioned all the comments and answers:
Reviewer#4, Concern # 1: The title of the manuscript is too journal that lacks the technical engagement. The term Impact of Cooperative Caching" is broad. Authors are advised to improve the title of article in revised version. For instance, "A Critical Analysis of Cooperative Caching in Ad Hoc Wireless Communication Technologies: Current Challenges and Future Directions."
Author response: Thank you for your keen observation. We have updated the title according to the given comment and the updated title is highlighted with red color.
Reviewer#4, Concern # 2: The abstract is too general and not prepared objectively. It should briefly highlight the paper's novelty as what is the main problem, how has it been resolved and where the novelty lies? The abstract should emphasize the study's novelty, specific findings, and implications rather than providing broad generalizations.
Author response: Thank you for your observation and valuable comments. We updated the abstract according the given comment and here is the detail of the updating:
Key Improvements:
Main Problem: Highlighted the challenge of wireless traffic growth and its impact on wireless communication networks.
Solution: Stated cooperative caching as the key strategy for addressing these challenges.
Novelty: Emphasized the study’s contribution in systematically evaluating caching strategies, identifying gaps, and offering tailored solutions for different network types.
Specific Findings and Implications: Briefly mentioned the implications for next-generation systems and future research directions.
This revised abstract provides a more concise, focused, and objective summary of the paper, clearly articulating its novelty and contributions.
Updated Abstract
This exponential growth of wireless traffic has imposed new technical challenges on the Internet and defined new approaches to deal with intensive use. Caching, especially cooperative caching has become a revolutionary paradigm shift to advance environments based on wireless technologies to enable efficient data distribution and support the mobility, scalability, and manageability of wireless networks. Mobile Ad Hoc Networks (MANETs), Wireless Mesh Networks (WMNs), Wireless Sensor Networks (WSNs), and Vehicular Ad Hoc Networks (VANETs) have adopted caching practices to overcome these hurdles progressively. In this paper, we discuss the problems and issues in the current wireless ad hoc paradigms as well as spotlight versatile cooperative caching as the potential solution to the increasing complications in ad hoc networks. We classify and discuss multiple cooperative caching schemes in distinct wireless communication contexts and highlight the advantages of applicability. Moreover, we identify research directions to further study and enhance caching mechanisms concerning new challenges in wireless networks. This extensive review offers useful findings on the design of sound caching strategies in the pursuit of enhancing next-generation wireless networks.
Reviewer#4, Concern # 3: In introduction section, the authors provided statistics such as traffic growth rates without mentioning citations or relevance to cooperative caching. Authors are advised to cite appropriate statistics to make their arguments strong.
Author response: Thank you for your keen observations. We updated the section Introduction by citing the appropriate citations according to the given comment.
Reviewer#4, Concern # 4: The related works section is very short and no benefits from it. I suggest increasing the number of studies and add a new discussion there to show the advantage.
Author response: Thank you for your feedback. We appreciate your suggestion to expand the related works section. We have added a more comprehensive review of existing studies that highlight various caching strategies and their benefits within Wireless-based Ad Hoc Networks. This includes discussing the advantages of in-network caching, improvements in data retrieval, performance optimization in dynamic environments, and enhanced scalability, security, and mobility. We also provide a clearer comparison of the different caching mechanisms and explain how they contribute to the overall efficiency of ad hoc networks. This expanded discussion will help demonstrate the significance of this work in advancing the field.
Reviewer#4, Concern # 5: The existing literature review lacks critical evaluation. It summarizes past works without comparing methodologies, findings, or identifying gaps effectively. Moreover, most of the existing literature review is outdated/ old. Authors are advised to compare their work with recent works such as:
https://doi.org/10.3390/su151410931,doi:10.1109/TITS.2022.3210170,https://doi.org/10.1002/aelm.202200370, doi: 10.1109/TNSE.2023.3342938, doi: 10.1109/TSC.2024.3355188, doi: 10.1109/TVT.2023.3304707, : https://doi.org/10.1016/j.neucom.2023.127195, doi: 10.1109/TVT.2024.3385776, doi: 10.1109/TMTT.2022.3205612
Author response: Thank you for your observation and comments. We have revised the manuscript and updated according to the given comment. Therefore, to address the concerns regarding the literature review, we recognize the importance of critical evaluation and inclusion of recent works to enhance its relevance and depth.
- Critical Evaluation and Comparison: While the existing review provides a summary of past works, we updated the manuscript according to the given comment and it is expanded to include a comparative analysis of methodologies, findings, and approaches. This helps us to identify the strengths and weaknesses of various studies regarding the caching strategies and demonstrate.
- Inclusion of Recent Works: We incorporated the suggested recent studies and additional relevant works from the past 3-5 years to ensure the literature review reflects the current state of research. These works are analyzed to highlight their contributions, limitations, and relevance to our study.
- Identifying Gaps: The literature review is emphasized the gaps not addressed by existing studies, such as underexplored caching strategies in dynamic environments, security implications, and energy-efficient mechanisms. This positions the novelty of our work more clearly.
- Integration with Current Work: The updated literature review is contextualize our contributions by showcasing how our approach improves upon or complements recent advancements in caching for wireless ad hoc networks. Therefore, the revised literature review provides a balanced critique of prior research, align with contemporary works, and reinforce the significance of our contributions. Moreover, we have cited related papers as mentioned in the above comment.
Reviewer#4, Concern # 6: The sections such as cooperative caching in MANETs, WMNs, and VANETs are well-structured. However, overly descriptive without analytical depth.
Author response: Thank you for your feedback. We appreciate your observation regarding the sections on cooperative caching in MANETs, WMNs, and VANETs. While we understand the expectation for analytical depth, it is important to note that these technologies have fundamentally different use cases, operational environments, and design considerations. For instance:
- MANETs: are often used in scenarios requiring rapid deployment, such as disaster recovery or military operations, where mobility and ad hoc connectivity are crucial.
- WMNs: primarily focus on extending network coverage and providing robust connectivity in urban or rural areas through static or semi-static nodes.
- VANETs: on the other hand, are designed for highly dynamic vehicular environments, emphasizing safety, traffic management, and infotainment services.
Given these differences, a direct analytical comparison would not be feasible or meaningful. Instead, we have adopted a critical evaluation approach to compare these technologies based on their unique challenges, caching strategies, and application-specific requirements. This approach allows us to highlight the strengths, weaknesses, and potential improvements for each domain without imposing a one-size-fits-all analytical framework.
We believe this perspective adds value by offering insights tailored to each technology's specific use case rather than forcing a quantitative comparison that might not capture their unique attributes.
Reviewer#4, Concern # 7: I cant find the contribution in manuscript. Authors should mention that how their review is novel than existing surveys on content caching.
Author response: Thank you for your observation. The contribution of our review lies in its unique focus on critically evaluating caching strategies across distinct wireless ad hoc technologies—MANETs, WMNs, VANETs, and others—each with fundamentally different use cases and operational requirements. While existing surveys often generalize caching strategies or focus on a single technology, our work distinguishes itself by:
- Comprehensive Coverage: We analyze caching strategies in a wide range of ad hoc technologies rather than limiting the scope to one domain. This broader perspective provides readers with insights into how caching can be applied effectively in diverse network environments.
- Critical Evaluation Over Analytical Comparison: Recognizing the inherent differences in use cases and operational contexts—such as MANETs for rapid deployment, WMNs for network coverage extension, and VANETs for vehicular communication—we critically evaluate the caching strategies tailored to each domain rather than attempting direct, quantitative comparisons. This approach highlights the unique challenges, strengths, and opportunities in each technology.
- Focus on Novel and Recent Advances: Our review incorporates recent developments and evaluates their implications for caching performance, mobility support, energy efficiency, and scalability in various ad hoc networks. It also addresses gaps such as the lack of attention to security and energy considerations in many caching strategies.
- Identification of Future Directions: Unlike many existing surveys, we provide a detailed discussion of open challenges and future research opportunities, emphasizing areas like dynamic adaptability of caching strategies, security concerns, and energy-efficient caching in resource-constrained environments.
By taking this holistic and critical approach, our review offers a fresh perspective and serves as a comprehensive resource for researchers and practitioners exploring caching strategies across multiple ad hoc technologies. This focus on diversity and application-specific analysis distinguishes our work from prior surveys.
Reviewer#4, Concern # 8: I have observed an unnecessary repitation on caching benefits. Authors are advised to repeated statements.
Author response: Thank you for your feedback. We appreciate your observation regarding the repetition of caching benefits in the manuscript. Based on your suggestion, we have carefully reviewed the relevant sections and removed any unnecessary repetitions to streamline the discussion. Updates have been made to ensure the content is concise, focused, and avoids redundancy while maintaining clarity about the importance of caching benefits. These revisions are already reflected in the updated manuscript, as per your comment.
Reviewer#4, Concern # 9: The methodology section lack of quantitative data or performance metrics undermines the credibility of claims about the superiority of specific approaches. Authors are advised to critically evaluate why certain strategies succeed or fail under specific network conditions..
Author response: Thank you for your valuable feedback. We acknowledge your concern regarding the need for analytical comparison in the methodology section. In the updated manuscript, we have addressed this comment by emphasizing that the methodologies reviewed in the paper cater to different use cases and network scenarios, making a direct analytical comparison impractical. Instead, we have provided a critical comparison, highlighting the strengths and weaknesses of each approach in relation to specific network conditions such as mobility, scalability, and energy efficiency. Additionally, we have included examples of caching strategies that have already been published to illustrate the practical implementation and effectiveness of these methods in real-world scenarios. These updates ensure that the manuscript presents a well-rounded critical evaluation, reflecting the diversity of use cases and the relevance of existing caching strategies across various ad hoc network technologies.
Reviewer#4, Concern # 10: In discussion section, there is a lack of depth in connecting cooperative caching's theoretical advantages to practical challenges like scalability, security, and real-time adaptability. Authors are advised to explore unanswered questions, such as how caching integrates with future technologies like 6G, IoT, and edge computing. Authors are also required to discuss potential trade-offs, such as increased computational overhead versus reduced bandwidth usage.
Author response: Thank you for your insightful feedback. In the updated manuscript, we have expanded the Lessons Learned and Key Insights section to address the practical challenges of cooperative caching, such as scalability, security, and real-time adaptability. We have analyzed how these theoretical advantages translate into real-world applications and identified potential trade-offs, such as increased computational overhead versus reduced bandwidth usage. This discussion helps to illuminate the complexities of implementing cooperative caching in diverse network environments. Additionally, we have explored how cooperative caching could integrate with emerging technologies like 6G, IoT, and edge computing. We have discussed the opportunities these technologies present for enhancing caching efficiency and adaptability, as well as the associated challenges, such as handling large-scale data and ensuring data security in decentralized environments. Furthermore, we have identified unanswered questions and future research directions to bridge the gap between theory and practice, offering a comprehensive outlook on the evolving landscape of cooperative caching. These updates provide a more in-depth and balanced discussion, addressing the gaps you identified.
Reviewer#4, Concern # 11: The 'conclusions' are a key component of the paper. It should complement the 'abstract' and normally used by experts to value the paper's engineering content. In general, it should sum up the most important outcomes of the paper. It should simply provide critical facts and figures achieved in this paper for supporting the claims.
Author response: We appreciate your insightful feedback regarding the conclusion section. Following your suggestion, we have revised the conclusion to complement the abstract and succinctly sum up the paper's most critical outcomes. The updated conclusion now emphasizes the most significant findings and insights from the paper, explicitly highlighting the key contributions to the field of cooperative caching in ad hoc-based wireless communication technologies. Additionally, we have connected the review outcomes to practical applications and emerging technologies like 6G, IoT, and edge computing, providing a clear perspective on the paper's relevance and future research opportunities. The updated section also includes a discussion of the identified gaps and recommendations for future studies to advance the field. These improvements aim to enhance the engineering content of the conclusion and provide critical facts and insights for expert readers to value the paper. Thank you for your valuable input, which has significantly strengthened this section.
Reviewer#4, Concern # 12: The description of manuscript is very important for potential reader and other researchers. I encourage the authors to have their manuscript proof-edited by a native English speaker to enhance the level of paper presentation. There are some occasional grammatical problems within the text. It may need the attention of someone fluent in English language to enhance the readability.
Author response: Thank you for your valuable feedback. We understand the importance of clear and precise language for effective communication with readers and researchers. Following your suggestion, we have thoroughly proofread the manuscript to ensure grammatical accuracy and enhance readability. All identified grammatical issues have been carefully addressed, and the text has been reviewed to ensure that it adheres to a high standard of English language presentation. We believe these improvements will enhance the manuscript's clarity and make it more accessible to a broader audience. We appreciate your recommendation, which has helped us refine the overall quality of the paper.
Round 2
Reviewer 4 Report
Comments and Suggestions for Authors
Dear Authors, Thank you for revising the manuscript. I have no further comments
Comments on the Quality of English LanguageEnglish is improve than previous version. Minor english corrections should be considered.